# Bacterial Cyclic Dinucleotides and the cGAS–cGAMP–STING Pathway: A Role in Periodontitis?

**DOI:** 10.3390/pathogens10060675

**Published:** 2021-05-30

**Authors:** Samira Elmanfi, Mustafa Yilmaz, Wilson W. S. Ong, Kofi S. Yeboah, Herman O. Sintim, Mervi Gürsoy, Eija Könönen, Ulvi K. Gürsoy

**Affiliations:** 1Department of Periodontology, Institute of Dentistry, University of Turku, 20520 Turku, Finland; samira.h.elmanfi@utu.fi (S.E.); mervi.gursoy@utu.fi (M.G.); eija.kononen@utu.fi (E.K.); 2Department of Periodontology, Faculty of Dentistry, Biruni University, 34010 Istanbul, Turkey; myilmaz@biruni.edu.tr; 3Department of Chemistry and Purdue Institute for Drug Discovery and Purdue Institute of Inflammation, Immunology and Infectious Disease, Purdue University, West Lafayette, Indiana, IN 47907, USA; ong32@purdue.edu (W.W.S.O.); yeboahk@purdue.edu (K.S.Y.); 4Oral Health Care, Welfare Division, City of Turku, 20520 Turku, Finland

**Keywords:** periodontitis, bacterial recognition, nucleic acid, cyclic dinucleotides, STING

## Abstract

Host cells can recognize cytosolic double-stranded DNAs and endogenous second messengers as cyclic dinucleotides—including c-di-GMP, c-di-AMP, and cGAMP—of invading microbes via the critical and essential innate immune signaling adaptor molecule known as STING. This recognition activates the innate immune system and leads to the production of Type I interferons and proinflammatory cytokines. In this review, we (1) focus on the possible role of bacterial cyclic dinucleotides and the STING/TBK1/IRF3 pathway in the pathogenesis of periodontal disease and the regulation of periodontal immune response, and (2) review and discuss activators and inhibitors of the STING pathway as immune response regulators and their potential utility in the treatment of periodontitis. PubMed/Medline, Scopus, and Web of Science were searched with the terms “STING”, “TBK 1”, “IRF3”, and “cGAS”—alone, or together with “periodontitis”. Current studies produced evidence for using STING-pathway-targeting molecules as part of anticancer therapy, and as vaccine adjuvants against microbial infections; however, the role of the STING/TBK1/IRF3 pathway in periodontal disease pathogenesis is still undiscovered. Understanding the stimulation of the innate immune response by cyclic dinucleotides opens a new approach to host modulation therapies in periodontology.

## 1. Introduction

Periodontitis is a chronic inflammatory disease of the tooth-supporting tissues, and is characterized by progressive and irreversible breakdown of the periodontal ligament and the surrounding alveolar bone. The main features of the disease are attachment loss—which refers to the apical migration of the dentogingival junction—and bone resorption, leading to pocket formation (pathologically deepened gingival crevice) and increased tooth mobility (Figure 1A,B) [1].

In the oral cavity, host cells recognize bacteria and their byproducts via their pattern recognition receptors, i.e., toll-like receptors (TLRs), nucleotide-binding oligomerization domains, or peptidoglycan recognition proteins. These receptors are capable of regulating the innate immune response by promoting the production of pro-inflammatory cytokines and chemokines [2]. With the increase in tissue chemokine concentration and vascular permeability, neutrophils and monocytes migrate to the connective tissue. This early response is then followed by the migration of plasma cells, T and B lymphocytes, and macrophages. Increased numbers of phagocytic cells elevate the local proteolytic activity, which disrupts the integrity of the gingival connective tissue matrix. This inflammatory stage is still reversible, and may never progress to periodontitis. Notably, periodontitis is not developed in all individuals, even with poor oral hygiene, but a susceptible host is essential. When the inflammatory process expands deeper into the alveolar bone, leading to osteoclastogenesis and bone destruction, the disease is called periodontitis. Periodontal lesions with loss of attachment and alveolar bone are characterized by a dense infiltrate consisting mainly of plasma cells and macrophages [3,4,5].

Pathogenesis of periodontitis has been summarized in several well-written descriptive or narrative reviews [1,2,3,4,5]. The aim of the present narrative review is to evaluate the possible contributions of bacterial cyclic dinucleotides and their host cell receptors’ STING pathways to the pathogenesis of periodontal disease. The final part of the review will discuss the use of small molecule inhibitors and activators of the STING pathway as immune response regulators. The literature was searched using the terms “STING”, “STING and periodontitis”, “TBK1”, “TBK 1 and periodontitis”, “IRF3”, “IRF3 and periodontitis”, “cGAS”, and “cGAS and periodontitis” by S.E. and M.Y., using the databases of PubMed/Medline, Scopus, and Web of Science. Due to the nature of this review’s length, viral activation of the STING pathway and the role of viruses in periodontal disease pathogenesis were not covered by this review. Moreover, the present review only focused on the mammalian STING pathway.

## 2. Intracellular Nucleic Acid Receptors in Mammalian Cells and the STING/TBK1/IRF3 Pathway

The exposure of oral tissues to microbes is a continuous process. The release of extracellular DNA from microbial biofilms and damaged host cells during disease results in an increase in extracellular DNA in the oral cavity [6]. Extracellular DNA, as well as endocytosed/phagocytosed sources of nucleic acids (i.e., microbes, apoptotic/necrotic cells, etc.), are recognized by various pattern recognition receptors, including TLRs, RIG-I-like receptors (RLRs), and other cytosolic DNA sensors [7,8]. Indeed, these receptors may cooperatively work to protect the host from microbial invasion. Host cells can recognize the pathogen-associated proteins and nucleic acids (i.e., aberrant 5′ triphosphorylated or 5′ dephosphorylated double-stranded RNAs or DNAs, RNA–DNA hybrids, or cyclic dinucleotides of invading microbes) via surface-exposed and intracellular immune receptors [7,8]. This recognition stimulates the innate immune system to initiate its defense mechanisms. Any variation in the quantity and quality of nucleic acids can be sensed by highly systematized and actively maintained nucleic acid receptors [9]. 

Receptors involved in nucleic acid recognition can be generally divided into two groups based on cellular localization. The first group is composed of TLRs (especially TLR3, TLR7, TLR8, and TLR9), which are localized in the endoplasmic reticulum (ER), lysosomes, and endosome [10]. These are transmembrane receptors expressed in immune cells—including monocytes, macrophages, dendritic cells, and B cells—as well as in non-immune cells, such as keratinocytes and epithelial cells [11]. TLR activation triggers an immune response by a cascade of events: production of cytokines, stimulation of the major histocompatibility complex, and activation of other costimulatory molecules [12]. Clinical studies have demonstrated that TLR9 gene and protein expression are increased in gingival tissues associated with periodontitis [13]. Indeed, specific polymorphisms were observed in the promoter region of the TLR9 gene in individuals with periodontitis [14]. Of the pattern recognition receptors, TLR2 and TLR4 are extracellular receptors that can recognize various pathogen-associated molecular patterns (PAMPs) [15]. The second group consists of TLR-independent pathways, which are more varied in their composition and signaling, and detect nucleic acids in the cytoplasm. TLR-independent pathways include nucleotide-binding domain and leucine-rich-repeat-containing receptors, or NOD-like receptors (NLRs), RNA polymerase III, RLRs, retinoic acid-inducible gene I (RIG-I), melanoma differentiation-associated gene 5 (MDA5), mitochondrial antiviral signaling protein (MAVS), absent in melanoma 2 (AIM2), cyclic guanosine monophosphate (GMP)–adenosine monophosphate (AMP) (cGAMP) synthase (cGAS), and stimulator of IFN genes (STING) [16,17,18,19,20]. Among these receptors, the expression of AIM2 (cytosolic DNA sensor) was found to be elevated in periodontitis lesions in comparison to healthy gingival tissues [21,22]. A significant elevation in AIM2 expression was demonstrated in gingival fibroblasts in response to oral biofilm bacteria [23]. 

Physiological responses in a living organism require simultaneous interactions of various microbial products with diverse receptors [24]. STING is a critical and essential innate immune signaling adaptor molecule, which detects exogenous cytosolic double-stranded DNAs (dsDNA) or endogenous sources such as cyclic dinucleotides that have escaped DNase degradation, leading to the production of IFNs [25,26,27,28]. As the DNAs of most microorganisms (except for RNA viruses) and CDNs are considered to be PAMPs, sensing of cyclic dinucleotides by STING connects microbial cytosolic sensing with host cell activation, and gives STING a key role in host immune response [29,30]. Beyond the antimicrobial functions of cGAS and STING, recent evidence has expanded their roles to cancer, including other cellular functions such as DNA repair and autophagy [31]. The presence of cytosolic DNA activates cGAS, which is dependent on dsDNA length, whereas its absence will maintain cGAS in an unactivated state. The presence of dsDNA in the cytoplasm, and its binding with cGAS, cause a conformational change in cGAS by inducing the formation of liquid-like droplets and cytoplasmic cGAS–DNA foci. In the early phase, as part of a dynamic internal rearrangement, cGAS and DNA foci become mobile within liquid droplets. In the later phase, the liquid droplets mature to a gel-like state. GTP and ATP convert this binding to the endogenous second messenger named 2′3′-cGAMP [32,33,34,35,36]. 2′3′-cGAMP has a unique structure: It contains unusual mixed phosphodiester linkages between a 2′-hydroxyl group of GMP and a 5′-phosphate group of AMP, and between a 3′-hydroxyl group of AMP and a 5′-phosphate group of GMP, forming a novel 2′3′-cGAMP isomer [26]. This isoform functions as a second messenger to bind and activate its endoplasmic reticulum membrane-situated adaptor protein (STING) [20]. Upon activation, STING translocates to intermediate compartments between the endoplasmic reticulum and the Golgi apparatus [7]. During translocation, the carboxyl terminus of STING can activate the protein kinases, TANK-binding kinase 1 (TBK1), and IκB kinase (IKK), followed by phosphorylation of the transcription factor IFN regulatory factor 3 (IRF3) and the nuclear factor kappa B (NF-κB) inhibitor IκBα [26]. IKK activates the NF-κB by phosphorylation of inhibitory IκB, which is associated with NF-κB in the cytoplasm of resting cells. After phosphorylation, inhibitory IκB will be degraded by the proteasome, resulting in a release of NF-κB transcription factor subunits [26]. Then, NF-κB translocates together with phosphorylated IRF3 into the nucleus, in order to provide a synergistic response against invading pathogens. This translocation activates the transcription and expression of genes encoding type I IFNs (e.g., IFN-β) and various cytokines and chemokines (e.g., IL-6 and TNF) [37]. STING-activated Type I IFNs are key cytokines induced by antimicrobial and antiviral immunity. In order to avoid continuous innate immune-related pro-inflammatory cytokine expression, STING is controlled by negative feedback and rapidly subjected to degradation [29,38]. This makes the STING pathway an important regulator of host defense against pathogens, in addition to its essential role in protecting the host tissues from the development of cancer [28,30]. 

To the best of our knowledge, localizations or activations of STING pathway proteins have not been investigated in periodontal tissues. Our immunohistochemical analyses indicate that, in healthy gingiva (sulcular epithelium and underlying connective tissues), STING is weakly present in the epithelium and in the connective tissue. In periodontitis, however, a strong STING accumulation is detected in the basal epithelium (i.e., epithelium–connective tissue interface) and around vessel walls in the connective tissue. TBK1 was prominent in both healthy and inflamed epithelial and connective tissues, but in contrast to STING, it was weakly visible in the basal layers of the epithelium. Moreover, IRF3 accumulated close to the basement membrane in the gingival tissues of periodontitis patients (Figure 2).

## 3. Bacterial Cyclic Dinucleotides, the cGAS–cGAMP–STING Pathway, and Periodontitis

Cyclic dinucleotides, including cyclic di-guanosine monophosphate (c-di-GMP), cyclic di-adenosine monophosphate (c-di-AMP), and cyclic GMP–AMP (cGAMP), are major secondary signaling molecules in bacteria. The heterocyclic configuration of cyclic dinucleotides contains two bases of guanine or adenine bonded to ribose and phosphate groups involved in the formation of a phosphodiester bond. This bond connects the C3′ of one pentose ring with the C5′ of another to produce a 3′-5′ cyclic dinucleotide [40,41]. Their chemical structure is similar, but there is a specificity in the synthesis and degradation enzymes that are involved in different cyclic dinucleotides. Their synthesis is regulated by cGAS/DncV-like nucleotidyltransferases (CD-NTases) that can catalyze synthesis-diverse nucleotide signals. Bacterial CD-NTases form a family of signaling enzymes encompassing at least seven different protein groups, including the dinucleotide cyclase and its metazoan homolog cyclic GMP–AMP synthase (cGAS) [41]. c-di-GMP is synthesized by diguanylate cyclases, and c-di-AMP by adenylate cyclases, while both of them are degraded by phosphodiesterases (PDEs) [42]. 

c-di-GMP is a universal bacterial secondary messenger in Gram-negative bacteria, first identified in 1987 [43]. c-di-GMP acts as an intracellular signaling molecule, and it coordinates various signaling networks including, but not limited to, the regulation of bacterial motility, exopolysaccharide synthesis, bacterial biofilm formation, bacterial adhesion, cell cycle progression and division, and stress survival, as well as the synthesis and secretion of virulence factors and pathogenesis [43]. c-di-AMP plays an essential role in regulatory processes, particularly in Gram-positive bacteria. These roles include cell-wall homeostasis, DNA repair, diverse gene expression, biofilm formation, sporulation, antibiotic resistance, and metabolism. Genomic analysis of some periodontal bacteria, e.g., *Treponema denticola* and *Selenomonas sputigena*, showed the presence of diguanylate cyclase domains and at least two c-di-GMP-binding proteins; moreover, synthesis of c-di-GMP in *Porphyromonas gingivalis*, known as a major periodontal pathogen, was detected [42]. 

Cyclic dinucleotides, including c-di-GMP, c-di-AMP, and cGAMP, bind and stimulate the STING pathway [44]. The occupation of compact conformation of cyclic dinucleotides in both bacterial and human STING is similar [45]. However, because of unique polar contacts and extensive interactions between mammalian STING and 2′3′-cGAMP, mammalian STING binds 2′3′-cGAMP with a higher affinity than bacterial cyclic dinucleotides [44]. In contrast, the bacterial STING pathway, which has a defensive role against bacteriophages, prefers canonical 3′–5′-linked cyclic dinucleotides, but not human 2′3′-cGAMP. The R232-equivalent position in bacterial STING R151 is outwards flipping, and does not contact the cyclic dinucleotide backbone. Moreover, the location of a universally conserved T173 residue in bacterial STING is under the cyclic dinucleotide binding pocket, which minimizes the occupied place for ′-OH within 2′,3′-cGAMP [45,46]. Bacterial STING pathways are not covered in the current review; more details on this topic are reviewed in Morehouse, B.R, et al. 2020 and Millman, A, et al. 2020 [45,46]. STING consists of an N-terminal four-pass transmembrane domain followed by a cytosolic cyclic dinucleotide-binding domain and a loosely structured C-terminal tail (CTT). In a resting state, STING is kept autoinhibited by its CTTs. After activation, exposing a polymerization interface that triggers the formation of STING homo-oligomers rereleases CTTs of STING and increases the stability of STING dimers by intermolecular disulfide bonds [47]. 

Activation of periodontal cellular responses (i.e., keratinocytes, fibroblasts, and macrophages) by cyclic dinucleotides is presented in Figure 3. We have demonstrated that, in human gingival keratinocytes, c-di-AMP significantly elevates the expression levels of IL-1β, IL-1Ra, monocyte chemoattractant protein, and vascular endothelial growth factor, and neutralizes the effects of LPS on IL-8 response [48]. We recently demonstrated the contribution of cyclic dinucleotides to the *P. gingivalis* LPS regulation of the human gingival fibroblast response [49]. Global proteomics studies by our groups revealed that cyclic dinucleotides have differential activations of various pathways, indicating that non-STING pathways may also contribute to immune responses [50,51]. Furthermore, the observed relation between the phagocytic behavior of macrophages and the STING pathway suggests that bacterial cyclic dinucleotides could play a role in stimulating the elimination of bacteria by host cells.

Immune cells, such as polymorphonuclear neutrophils, macrophages, and dendritic cells are commonly found even in clinically healthy periodontal tissues [52,53]. Dendritic cells and macrophages have various TLRs with which to recognize pathogen-associated molecular patterns, such as the LPS or flagellin of pathogenic bacteria [53,54,55]. Activated dendritic cells induce local inflammation in order to eliminate the challenge by periodontal pathogens and to repair damaged periodontal tissues [53]. Stimulation of dendritic cells by *P. gingivalis* and its LPS leads to elevated expressions of TNF-α and IL-8 [56,57]. Neutrophils and monocytes are attracted to sites of inflammation by chemokines’ production of dendritic cells, which also stimulate the inflammation and expression of factors connected to osteoclastogenesis [58,59]. Macrophages are critical cells for the immunity and inflammatory processes. In inflammation, macrophages either limit the infectious process and complete healing with fibrosis and scar tissue formation, or they fail to clear the infection, leading to a chronic inflammatory lesion [60]. Cytoplasmic DNA can be recognized in macrophages and dendritic cells via STING in cooperation with cGAS to activate the TBK1/IRF3 and NF-κB pathways and to produce IFNs and TNF, respectively [61,62]. 

## 4. Small Molecule Modulators of the STING Pathway

Thus far, most efforts towards the development of STING pathway modulators have been directed towards anticancer therapy [27,31,63]. Inhibitors of STING signaling are being developed for inflammatory diseases, such as Aicardi–Goutières syndrome, STING-associated vasculopathy with onset in infancy, and systemic lupus erythematosus [30,63], but the research is limited. Mammalian STING pathway modulators, including the cyclic dinucleotide analogs, should be tested in the treatment of periodontitis as host-modulating adjuvants.

## 5. Inhibitors of STING Activation

A cyclopentapeptide, Astin C, was reported as being a competitive antagonist to inhibit the cGAS–STING signaling pathway, suppressing the innate inflammatory response [64]. The binding affinity of Astin C was determined using the ITC (isothermal titration calorimetry)-based assay to be 53 nM for STINGR232, comparable to the 50 nM binding affinity for cGAMP [64,65]. Astin C dampens STING-dependent IFN-β activation, with an IC_50_ of 10.83 ± 1.88 and 3.42 ± 0.13 µM for human and murine fibroblasts, respectively. Furthermore, Astin C attenuated autoinflammatory responses and innate antimicrobial defenses, both in vitro and in vivo [64]. 

TBK1 activation has been recently found to be due to palmitoylation of Cys88 and Cys91 in STING [66]. Palmitoylation of proteins regulates protein trafficking and stability by disrupting interactions with lipids and other membrane-bound proteins. Palmitoylation of the TM2–TM3 linker in STING contributes to the tetramer interface, and the fatty acid label is conjugated on STING from the Golgi apparatus until STING is degraded [66]. These illustrations suggest that the palmitoylation of STING is a promising therapeutic target for STING signaling activation. Nitrofurans and nitro-fatty acids (NO2-FAs) have been shown to be potent inhibitors of STING [67,68]. The mode of action of these classes of compounds is by inhibiting the activation-induced palmitoylation of STING. NO2FA (10-nitrooleate, 9-nitroleate, and nitro-conjugated linoleic acid) and nitrofuran compounds (C-170, C-176, and C-178) are suggested to prevent STING signaling by a similar mechanism of action: alkylation of cysteine [69]. Nitrofurans C-176 and C-178 (Figure 4) were initially discovered as covalent inhibitors of STING via an IFN-β luciferase reporter assay [68]. Structure–activity relationship (SAR) studies showed that both the furan and nitro groups are essential for inhibition, and electron withdrawing groups such as -CF3 and -Br at position 4 of the phenyl ring increased inhibition potency. Interestingly, the incorporation of a methyl group at the main amide nitrogen of C-176 completely removes the compound’s inhibitory ability. Though the half-life of C-173 is 1 h in mice, its inhibitory capability is still potent, reducing STING-associated autoinflammation in CMA-treated and Trex1-/- mice without toxicity [68]. C-176 and C-178 only inhibit mSTING and not hSTING. Subsequent modification afforded analogs (C-170 and C-171) that effectively inhibit both mSTING and hSTING through the same covalent modification. Additional screening led to the discovery of H-151, which covalently binds to Cys91 in STING. Pretreatment with H-151 dramatically suppresses cytokine levels in CMA-treated and Trex1-/- mice [68].

## 6. Activators of STING Pathway

While the STING pathway can be activated through STING binding to cyclic dinucleotides such as c-di-GMP, c-di-AMP, 2′-3′cGAMP, and 3′-3′cGAMP [70], key challenges that limit the direct utility of these cyclic dinucleotides as drug agonists include high negative charge, hydrophilicity, and propensity towards degradation of cyclic dinucleotides by phosphodiesterases such as ENPP1 [71,72]. Possible strategies that have been employed to overcome such limitations to date include the design of synthetic cyclic dinucleotide analogues [73] and other non-nucleotide-based STING agonists [74,75]. In 2015, Gejewski et al. synthesized and investigated the binding affinities of various cyclic dinucleotide analogues across both human and murine STING alleles [71]. ADU-S100, a (R,R) bisphosphorothioate analogue of c-di-AMP, was found to exhibit enhanced stability and improved STING activation effects compared to non-modified cyclic dinucleotides such as c-di-AMP, ci-di-GMP, and cGAMP. In vitro treatment of ADU-S100 in WT C57BL/6 and STING^(−/−)^ murine bone marrow macrophages showed significantly higher expression of interferon-β when compared to stimulation using endogenous PAMPs, such as c-di-GMP and 2′-3′cGAMP. Furthermore, in vivo experiments utilizing ADU-S100 treatment on mice bearing B16F10 tumors showed significant tumor growth compared with negative control and 2′-3′cGAMP treatments. ADU-S100 currently remains in phase II clinical trials in combination with pembrolizumab for the treatment of neck and head cancer (NCT03937141), with preliminary evidence suggesting generally good tolerance of this drug combination [76]. The development of analogues such as ADU-S100 has also spurred the optimization of next-generation cyclic dinucleotide analogues as STING agonists, such as E7766 [77]. E7766 is a macrocycle-bridged cyclic dinucleotide STING agonist (Figure 5) developed by Eisai Inc., which was shown to increase INF-β expression in a dose-dependent manner in ex vivo experiments utilizing human primary peripheral blood mononuclear cells (PMBCs). In vivo experiments using a CT26 dual-tumor model in BALB/cJ mice have also revealed that direct administration of E7766 to the tumor results in a dose-dependent reduction in tumor volume. Currently, E7766 is undergoing phase I clinical trials for the treatment of lymphomas as of 2021 (NCT04144140). 

High-throughput screening efforts have also led to the identification of small-molecule STING agonists. As early as 2015, Sali et al. identified the STING agonist G10 (Figure 5) through an IRF3-activation-based high-throughput screen [78]. G10 was shown to selectively activate INF/IRF3 signaling. While it was postulated that the agonistic effects may not be due to the direct activation of STING by G10, in vitro experiments utilizing the stimulation of HEK293T harboring a luciferase reporter expressed with an ISRE/ISG promoter using G10 confirmed G10 to be a weak direct binder of STING, with differing potencies to the STING variant expressed [79].

In 2019, Ramanjulu et al. also reported the design of the dimeric ligand diABZI compound 3 as a STING agonist (Figure 5) [80]. In vitro studies with human PMBCs revealed the dose-dependent induction of INF-β upon diABZI compound 3 treatment, with an apparent EC50 of 130 nM. Compound 3 was also shown to exhibit potent antitumor effects in in vivo experiments using a CT-26 tumor model in BALB/c mice, where tumor growth inhibition and improved survival were noticed on an intermittent dosing scheme.

Chin et al. also recently discovered the small-molecule STING agonist SR-717 (Figure 5) [74]. Crystal structures of hSTING-SR-717 revealed that SR-717 is a cGAMP mimetic capable of promoting STING activation through the establishment of a closed conformation when bound to hSTING. Treatment of SR-717 in THP-1 and PMBC cell lines also showed a dose-dependent response in the activation of IRF3 and INF-β signaling through in vitro experiments. The role of SR-717 as an agonist was further corroborated via in vivo experiments, where intraperitoneal administration of SR-717 resulted in dose-dependent elevation of INF-β levels, while INF-β levels in Sting^gt/gt^ mutant variants remained unaffected upon SR-717 treatment. Recently, the orally available STING agonist MSA-2 was identified, bearing high therapeutic efficacy [75]. While MSA-2 was shown to be unable to bind to STING, it was also shown to form non-covalent dimers in solution, which bind extensively to STING; characterization of a surrogate covalent dimer of MSA-2 demonstrated its high binding affinity towards WT STING, with an IC_50_ of 23 ± 7nM. In vitro studies have also confirmed a dose-dependent induction of INF-β in THP-1 and murine macrophage models, which was absent when STING^−/−^ THP-1 cells were administered with MSA-1. Furthermore, MSA-2 exhibited potent dose-dependent antitumor activity, with tumor regressions in 80–100% of the animals in an MC38 syngeneic murine model. 

## 7. Conclusions

Here, we present the available evidence on the role of bacterial secondary signaling molecules—cyclic dinucleotides—in the regulation of periodontal immune response via the STING/TBK1/IRF3 pathway. We also demonstrate evidence that this pathway can be inhibited or activated by small molecules. Considering that mainly commensal bacteria, and only a handful of pathogenic species, reside in the oral cavity, the perplexing question remains: how does the innate immune system differentiate the DNA and cyclic dinucleotides of commensal bacteria from those of pathogenic species? The central role of cyclic dinucleotides in the stimulation of innate immune response to bacterial and viral infections offers a new approach to regulating prokaryotic and eukaryotic innate immune systems [81,82].

## Figures and Tables

**Figure 1 pathogens-10-00675-f001:**
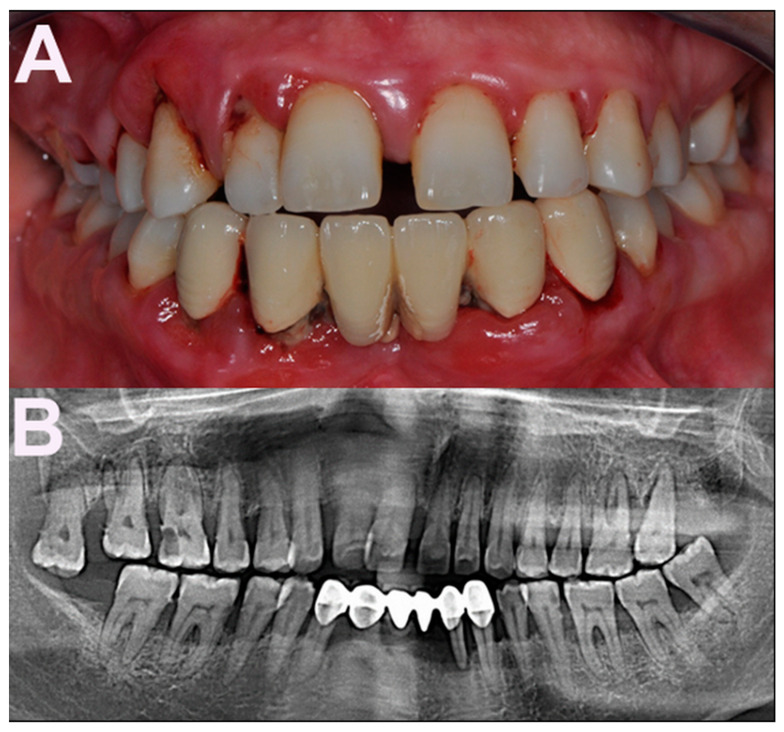
(**A**) Clinical appearance of advanced periodontitis in a 32-year-old, otherwise healthy, female patient. Heavy dental biofilm deposits and signs of gingival inflammation are clinically visible. (**B**) Panoramic radiograph of the patient reveals generalized alveolar bone destruction and its severity (by courtesy of Mustafa Yilmaz, Biruni University, Istanbul, Turkey).

**Figure 2 pathogens-10-00675-f002:**
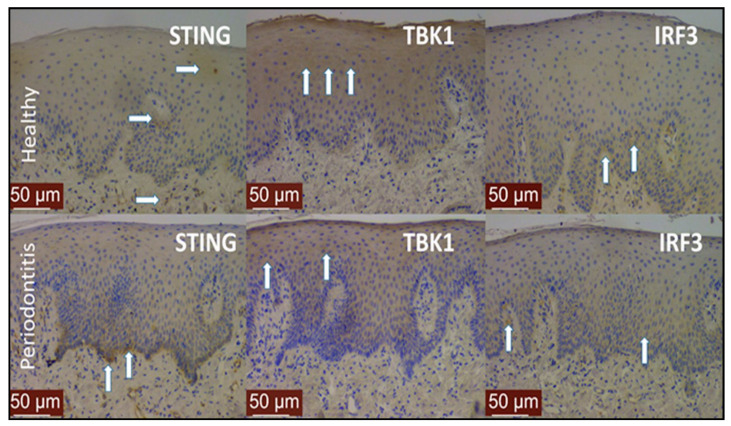
Localizations of STING, TBK1, and IRF3 in gingival tissues with good periodontal health (**upper row**) and with periodontitis (**lower row**). White arrows indicate examples of zones with prominent staining. Periodontally healthy gingival tissue was excised during a crown lengthening procedure from a 30-year-old female. The gingival tissue sample with periodontitis was excised during a flap surgery. Both surgical interventions were part of the respective patients’ routine periodontal treatment, and the samples belong to the sample collection of Dr. Gökhan Kasnak, Cerrahpasa University, Istanbul, with the ethical permission no: Istanbul University 2017/41. Histological techniques were performed according to the previously described method [39], using primary antibodies of STING (PA5-26808, Thermo Fisher, Rockford, IL, USA), IRF3 (PA5-87506, Thermo Fisher), and TBK1 (PA5-17478, Thermo Fisher).

**Figure 3 pathogens-10-00675-f003:**
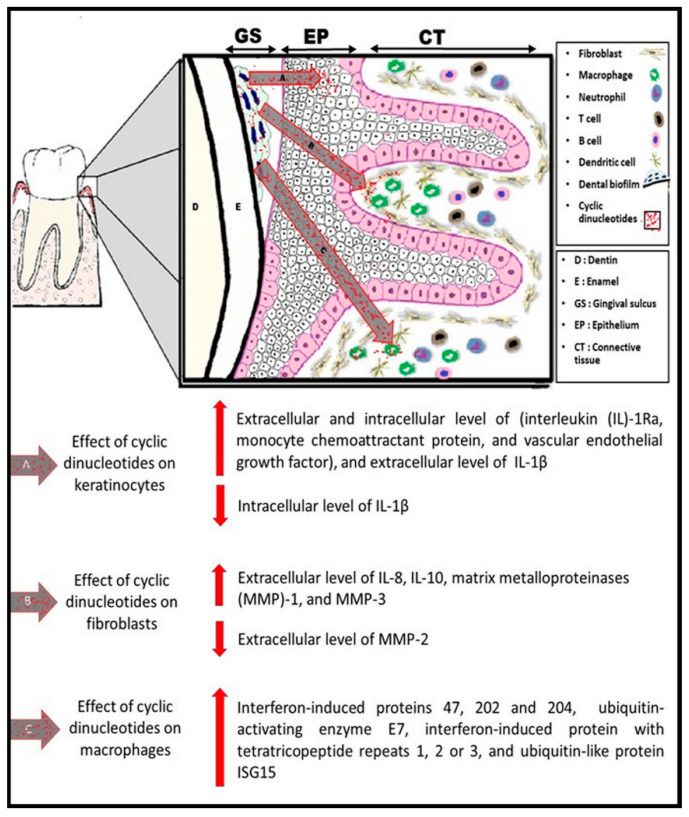
Diagram illustrating the cellular response of three types of periodontal resident cells (keratinocytes, fibroblasts, and macrophages) to bacterial cyclic dinucleotides.

**Figure 4 pathogens-10-00675-f004:**
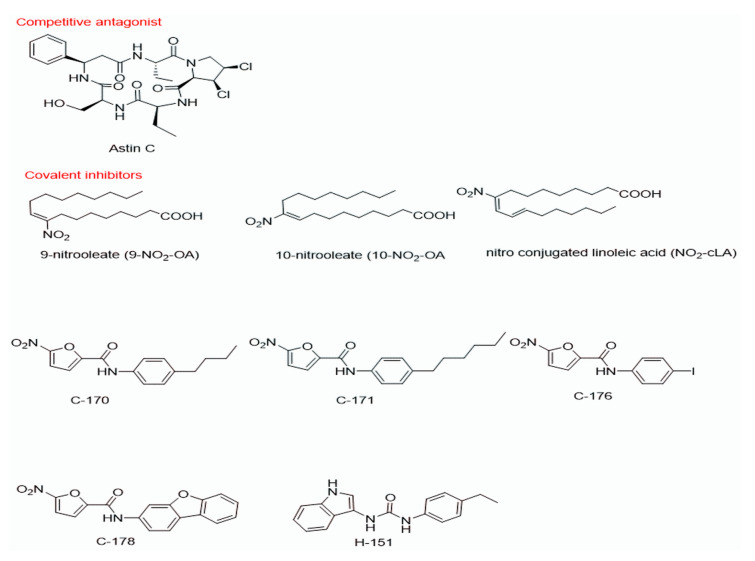
Selected STING inhibitors.

**Figure 5 pathogens-10-00675-f005:**
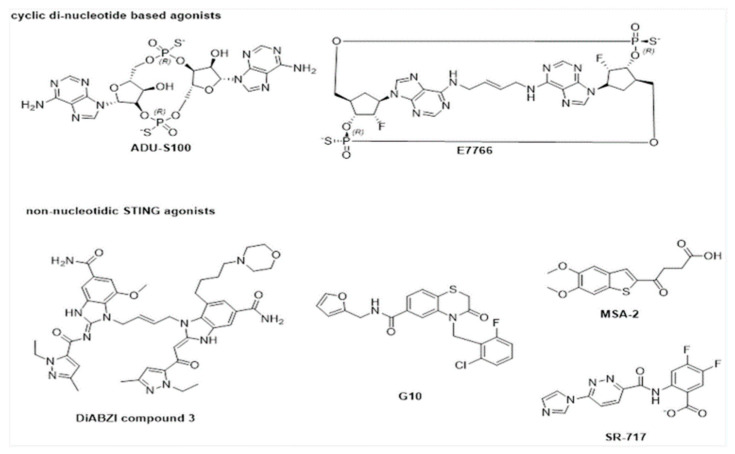
Chemical structures of STING agonists.

## Data Availability

The study did not report any data.

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
