# Peer review of "Bacterial Cyclic Dinucleotides and the cGAS–cGAMP–STING Pathway: A Role in Periodontitis?"

_pathogens, 2021, doi:10.3390/pathogens10060675_

Round 1

Reviewer 1 Report

The authors present a narrative review on the role of bacterial Cyclic Dinucleotides and cGAS–cGAMP–STING 2 in the pathogenesis of periodontitis. The manuscript is well written and contains important fundamental research outcomes. Some minor aspects would improve the paper:

Abstract:

  • Please mention the method and type of your review: is this a narrative or systematic review?
  • Please mention which databases were searched and how
  • Please mention clearly what were the results of this review.

Entire manuscript: please define the search strategy, search terms, searched databases, investigators, e.t.c.

Introduction:

  • Please reconsider the aim: what is a “theory-driven interpretative review”?. Please use the current classification of reviews and define this review accordingly. Moreover, please rephrase “to justify the possible contributions”- the aim is rather to determine, to evaluate e.t.c. but for sure not to “justify”.

2 Inracellular nucl. Acid receptors in mammalian cells:

  • Please indicate in what conditions with what technique were the hystologies performed. What regions were the biopsies taken from? What diagnosis had the patient from which the biopsies were taken from? Is there a signed written consent from the patient for this histological analysis and publication?
  • 2: please indicate exactly in the images the localizations of STING, TBKq, IRFbeta.

 Inhibitors and activators of STING

- please indicate the type of studies (in vivo, in vitro) where the STING inhibitors were developed. Are there any clinical reports on these?

Author Response

Referee 1:

We thank to referee for his positive and constructive comments. A response to each comment can be found below (underlined). Changes in the text are highlighted with yellow.

Abstract:

  • Please mention the method and type of your review: is this a narrative or systematic review?
  • Please mention which databases were searched and how
  • Please mention clearly what were the results of this review.

Authors’ response: PubMed/Medline, Scopus, and Web of science were searched with the terms “STING”, “TBK 1”, “IRF3”, “cGAS” alone and together with ”periodontitis”. Information is included into the text (page 1, line 26-27).

Entire manuscript: please define the search strategy, search terms, searched databases, investigators, e.t.c.

Authors’ response: The literature was searched using the terms “STING”, “STING and periodontitis”, “TBK1”, “TBK 1 and periodontitis”, “IRF3”, “IRF3 and periodontitis, and ”cGAS”, “cGAS and periodontitis” by S.E. and M.Y. from databases of PubMed/Medline, Scopus, and Web of Science. Information is included into the text (page 2, line 66-69).

Introduction:

  • Please reconsider the aim: what is a “theory-driven interpretative review”?. Please use the current classification of reviews and define this review accordingly. Moreover, please rephrase “to justify the possible contributions”- the aim is rather to determine, to evaluate e.t.c. but for sure not to “justify”.

Authors’ response: Aim sentence is corrected (page 2, line 62).

  1. Inracellular nucl. Acid receptors in mammalian cells:
  • 1. Please indicate in what conditions with what technique were the hystologies performed. What regions were the biopsies taken from? What diagnosis had the patient from which the biopsies were taken from? Is there a signed written consent from the patient for this histological analysis and publication?

Authors’ response: Detailed information on the source of samples and histopathological analyses are included into the text (page 5, line 164-172).

  • 2: please indicate exactly in the images the localizations of STING, TBKq, IRFbeta.

Authors’ response: Localizations of STING, TBK1, IRF 3 marked by arrows in the figure 2 (page 5).

 Inhibitors and activators of STING

- please indicate the type of studies (in vivo, in vitro) where the STING inhibitors were developed. Are there any clinical reports on these?

Authors’ response: In-vitro and in-vivo experiments for individual compounds in section 6 has been incorporated and elaborated into the section where applicable. It is noted that with the exception of ADU-S100 and E7766, the listed compounds lack clinical data from clinical trials to date. E7766 is currently undergoing phase I clinical trials, with no data revealed. For G10, in-vivo data was not mentioned as in-vivo data was not reported, information is incorporated into the text in section 6 (page 9 and 10, line 305-351).

Reviewer 2 Report

Reviewer comments:

  1. Incomplete sentence after semicolon. Suggestion is inserted in bold.
  • “In this review, we (1) focus on the possible role of bacterial cyclic 22 dinucleotides and STING/TBK1/IRF3 pathway in the pathogenesis of periodontal disease and regulation of periodontal immune response; we (2) review and discuss activators and inhibitors of STING pathway as immune response regulators and their potential utility in the treatment of periodontitis.”

  1. STING pathway studies in context of infection have been performed (PMID: 28625530; PMID: 31827069; PMID: 30001525, etc). This is phrased in such a way that a reader may conclude otherwise. Further, suggest using alternative word, rather than illustrated in the second sentence.
  • “Current studies have directed to using STING pathway activators as an anticancer therapy and using STING signaling inhibitors for inflammatory diseases. In contrast, cyclic dinucleotides' role in regulating periodontal immune response by the STING/TBK1/IRF3 pathway remains to be illustrated.”

  1. Edit this sentence for clarity
  • “Phagocytosis and the release of extracellular DNA from biofilms increase the abundancy of extracellular DNA in the oral cavity [6]. Nucleic acids from pathogens are recognized by various pattern recognition receptors, including TLRs, RIG-I-like receptors (RLRs), and other cytosolic DNA sensors.”

  • Example edit: The release of extracellular DNA from microbial biofilms and damaged host cells during disease results in an increase of extracellular DNA in the oral cavity. Extracellular DNA as well as endocytosed/phagocytosed sources of nucleic acids (i.e., microbes, apoptotic/necrotic cells) are recognized by various pattern recognition receptors, including TLRs, RIG-I-like receptors (RLRs), and other cytosolic DNA sensors.

  1. Edit punctuation: Missing parenthesis. Use brackets inside parentheses to create a double enclosure in the text.
  • “Host cells can recognize the pathogen-associated proteins and nucleic acids (i.e., aberrant (5ʹ triphosphorylated or 5ʹ diphosphorylated) double-stranded RNAs or DNAs, RNA–DNA hybrids, or cyclic dinucleotides of invading microbes by surface-exposed and intracellular immune receptors [7,8].”
  • Example: Host cells can recognize the pathogen-associated proteins and nucleic acids (i.e., aberrant [5ʹ triphosphorylated or 5ʹ dephosphorylated] double-stranded RNAs or DNAs, RNA–DNA hybrids, or cyclic dinucleotides of invading microbes by surface-exposed and intracellular immune receptors) [7,8].

  1. Awkward phrasing: Although not incorrect, I suggest using alternative phrasing, rather than acts as an alert.
  • “This recognition acts as an alert to the innate immune system to initiate the defense mechanisms.”
  • Examples: (“This recognition alerts the innate…”) (“This recognition acts as a signal to the innate…”) (“This recognition acts as an alert system to the innate…”)

  1. Edit this sentence for clarity and expand into a couple of sentences. Example of key facts to incorporate: DNA from exogenous or endogenous sources that have escaped DNase degradation are detected (PMID 29706455) and cGAS binding is dependent on dsDNA length (PMID 28902841)
  • “STING is a critical and essential innate immune signaling adaptor molecule, which detects cytosolic double strand DNAs (dsDNA) derived from microbes and endogenous second messengers such as cyclic dinucleotides, leading to the production of IFN [25,26].”

  1. Replace the word cell with host tissues. Also, it is clear you need to update your review with more recent data from studies of STING in context of infection such as PMID 29924997 (etc) and appropriate sources used in reviews PMID: 28625530 and PMID: 31827069.
  • “STING-activated Type I IFNs are key cytokines induced by antimicrobial and antiviral immunity. To avoid continuous innate immune-related pro-inflammatory cytokine expression, STING is rapidly subjected to degradation [32]. This makes the STING pathway an important regulator of host defense against pathogens besides its essential role in protecting the cell from the development of cancer [32].”

  1. Methodology is missing for Figure 2. Hard to determine what you are referring to in your description. What antibodies? What stains? What does purple and brown color mean. What am I looking for? Is this human tissue or infected animal? How was it prepared? Is this data linked to a previous study you published?
  • “TBK1 was prominent in both healthy and inflamed epithelial and connective tissues, but in contrast to STING, it was weakly visible in basal layers of the epithelium.”

  1. Edit for clarity, the whole section titled Bacterial Cyclic Dinucleotides, cGAS–cGAMP–STING Pathway, and Periodontitis. It is not always clear if you are talking about bacterial STING pathway and its components or mammalian STING pathway or both. Though the Human pathway is obviously the primary topic in this review, because you added this section, the fact that many bacterial species have a STING pathway, presumably used primarily as a defense against bacteriophage, you have to clarify.

  1. You should also expand on the distinctions of bacterial versus mammalian in section 3. For example, you wrote:
  • “Cyclic dinucleotides, including c-di-GMP, c-di-AMP, and cGAMP, bind and stimulate the STING pathway. However, as a result of unique polar contacts and extensive interactions between STING and 2′3′-cGAMP, STING binds 2ʹ3ʹ-cGAMP with a higher affinity than bacterial cyclic dinucleotides [37].”

That this sentence (above) refers to mammalian STING, which was not clear to me. In bacteria, STING prefers the conical 3’-5’ cdiGMP and are unable to bind to human 2’3’-cGAMP (PMID32877915).  Further, if the STING pathway is found in bacterial species with a CBASS system, it appears they are devoid of cdiAMP signaling domains suggesting that the normal role of cyclic dinucleotides in intracellular signaling to coordinate bacterial processes like biofilm formation has been diverted in order for STING to function in these bacterial species, because there is no way to tell if the source is endogenous or exogenous (bacteriophage).  (PMID32877915, PMID: 32839535)

According to PMID: 32877915, STING system originated in bacteria.

  1. Add specific examples to clarify. This sentence is OK but it could be better. I would argue periodontal disease is an inflammatory disease because it depends on the host and it is the host response cause the majority of the tissue destruction. It is not merely an infectious disease. What specific inflammatory diseases are you referring to here?
  • “Thus far, most efforts towards the development of activators of the STING pathway have been directed to anticancer therapy, while inhibitors of STING signaling are being developed for inflammatory diseases. Both activators and inhibitors of STING, described below or analogs thereof, could have potential utility in the treatment of periodontitis.”

  1. Figure 4 image quality/distortion. Recommend replacing with a better image.

  1. Awkward phrasing and unclear description. Although molecules may act as an agonist, using the term in the verb form “agonized” is odd because a molecule does not have feelings. Further, agonists act as inhibitors/repressors, but cyclic dinucleotides act as inducers of IFNs via the STING pathway. The paper you reference report it does this by degradation of STING proteins. The way you wrote it is very confusing. You should expand these points to make it clear what you mean by agonist activity.
  • “While STING is readily agonized through the binding of cyclic dinucleotides such as c-di-GMP, c-di-AMP, 2’-3’cGAMP and 3’-3’cGAMP [60], key challenges that limit their direct utility as drug agonists include high negative charge, hydrophilicity and propensity towards degradation of cyclic dinucleotides by phosphodiesterases such as ENPP1 [61,62].”

Author Response

Referee 2:

We thank to referee for his positive and constructive comments. A response to each comment can be found below (underlined). Changes in the text are highlighted with yellow.

Reviewer comments:

  1. Incomplete sentence after semicolon. Suggestion is inserted in bold.

“In this review, we (1) focus on the possible role of bacterial cyclic 22 dinucleotides and STING/TBK1/IRF3 pathway in the pathogenesis of periodontal disease and regulation of periodontal immune response; we (2) review and discuss activators and inhibitors of STING pathway as immune response regulators and their potential utility in the treatment of periodontitis.”

Authors’ response: Test is corrected according to the reviewer’s suggestion (page 1, line 24).

  1. STING pathway studies in context of infection have been performed (PMID: 28625530; PMID: 31827069; PMID: 30001525, etc). This is phrased in such a way that a reader may conclude otherwise. Further, suggest using alternative word, rather than illustrated in the second sentence.
  • “Current studies have directed to using STING pathway activators as an anticancer therapy and using STING signaling inhibitors for inflammatory diseases. In contrast, cyclic dinucleotides' role in regulating periodontal immune response by the STING/TBK1/IRF3 pathway remains to be illustrated.”

Authors’ response: Text is corrected according to the reviewer’s suggestion (page 1, line 27-31).

  1. Edit this sentence for clarity
  • “Phagocytosis and the release of extracellular DNA from biofilms increase the abundancy of extracellular DNA in the oral cavity [6]. Nucleic acids from pathogens are recognized by various pattern recognition receptors, including TLRs, RIG-I-like receptors (RLRs), and other cytosolic DNA sensors.”

  • Example edit: The release of extracellular DNA from microbial biofilms and damaged host cells during disease results in an increase of extracellular DNA in the oral cavity. Extracellular DNA as well as endocytosed/phagocytosed sources of nucleic acids (i.e., microbes, apoptotic/necrotic cells) are recognized by various pattern recognition receptors, including TLRs, RIG-I-like receptors (RLRs), and other cytosolic DNA sensors.

Authors’ response:  Text is corrected as reviewer suggested (page 3, line 75-80).

  1. Edit punctuation: Missing parenthesis. Use brackets inside parentheses to create a double enclosure in the text.
  • “Host cells can recognize the pathogen-associated proteins and nucleic acids (i.e., aberrant (5ʹ triphosphorylated or 5ʹ diphosphorylated) double-stranded RNAs or DNAs, RNA–DNA hybrids, or cyclic dinucleotides of invading microbes by surface-exposed and intracellular immune receptors [7,8].”
  • Example: Host cells can recognize the pathogen-associated proteins and nucleic acids (i.e., aberrant [5ʹ triphosphorylated or 5ʹ dephosphorylated] double-stranded RNAs or DNAs, RNA–DNA hybrids, or cyclic dinucleotides of invading microbes by surface-exposed and intracellular immune receptors) [7,8].

Authors’ response:  Text is corrected according to the reviewer’s suggestion (page 3, line 81-85).

  1. Awkward phrasing: Although not incorrect, I suggest using alternative phrasing, rather than acts as an alert.
  • “This recognition acts as an alert to the innate immune system to initiate the defense mechanisms.”
  • Examples: (“This recognition alerts the innate…”) (“This recognition acts as a signal to the innate…”) (“This recognition acts as an alert system to the innate…”)

Authors’ response: Text is rephrased (page 3, line 85).

  1. Edit this sentence for clarity and expand into a couple of sentences. Example of key facts to incorporate: DNA from exogenous or endogenous sources that have escaped DNase degradation are detected (PMID 29706455) and cGAS binding is dependent on dsDNA length (PMID 28902841)
  • “STING is a critical and essential innate immune signaling adaptor molecule, which detects cytosolic double strand DNAs (dsDNA) derived from microbes and endogenous second messengers such as cyclic dinucleotides, leading to the production of IFN [25,26].”

Authors’ response: Text is re-written (page 3 and 4, line 113-122).

  1. Replace the word cell with host tissues. Also, it is clear you need to update your review with more recent data from studies of STING in context of infection such as PMID 29924997 (etc) and appropriate sources used in reviews PMID: 28625530 and PMID: 31827069.
  • “STING-activated Type I IFNs are key cytokines induced by antimicrobial and antiviral immunity. To avoid continuous innate immune-related pro-inflammatory cytokine expression, STING is rapidly subjected to degradation [32]. This makes the STING pathway an important regulator of host defense against pathogens besides its essential role in protecting the cell from the development of cancer [32].”

 Authors’ response:  Text is corrected according to the reviewer’s suggestions (page 4, line 149).

  1. Methodology is missing for Figure 2. Hard to determine what you are referring to in your description. What antibodies? What stains? What does purple and brown color mean. What am I looking for? Is this human tissue or infected animal? How was it prepared? Is this data linked to a previous study you published?
  • “TBK1 was prominent in both healthy and inflamed epithelial and connective tissues, but in contrast to STING, it was weakly visible in basal layers of the epithelium.”

Authors’ response: A detailed description of methodology is implemented into the text (figure 2, legend (page 5).

  1. Edit for clarity, the whole section titled Bacterial Cyclic Dinucleotides, cGAS–cGAMP–STING Pathway, and Periodontitis. It is not always clear if you are talking about bacterial STING pathway and its components or mammalian STING pathway or both. Though the Human pathway is obviously the primary topic in this review, because you added this section, the fact that many bacterial species have a STING pathway, presumably used primarily as a defense against bacteriophage, you have to clarify.

Authors’ response: A description of the present review’s interest (mammalian STING) is included into the text (page 6, line 201-213).

  1. You should also expand on the distinctions of bacterial versus mammalian in section 3. For example, you wrote:
  • “Cyclic dinucleotides, including c-di-GMP, c-di-AMP, and cGAMP, bind and stimulate the STING pathway. However, as a result of unique polar contacts and extensive interactions between STING and 2′3′-cGAMP, STING binds 2ʹ3ʹ-cGAMP with a higher affinity than bacterial cyclic dinucleotides [37].”

That this sentence (above) refers to mammalian STING, which was not clear to me. In bacteria, STING prefers the conical 3’-5’ cdiGMP and are unable to bind to human 2’3’-cGAMP (PMID32877915).  Further, if the STING pathway is found in bacterial species with a CBASS system, it appears they are devoid of cdiAMP signaling domains suggesting that the normal role of cyclic dinucleotides in intracellular signaling to coordinate bacterial processes like biofilm formation has been diverted in order for STING to function in these bacterial species, because there is no way to tell if the source is endogenous or exogenous (bacteriophage).  (PMID32877915, PMID: 32839535)

According to PMID: 32877915, STING system originated in bacteria.

Authors’ response: In line to our response to comment 9, a chapter is implemented into the text (page 6, line 201-213).

  1. Add specific examples to clarify. This sentence is OK but it could be better. I would argue periodontal disease is an inflammatory disease because it depends on the host and it is the host response cause the majority of the tissue destruction. It is not merely an infectious disease. What specific inflammatory diseases are you referring to here?
  • “Thus far, most efforts towards the development of activators of the STING pathway have been directed to anticancer therapy, while inhibitors of STING signaling are being developed for inflammatory diseases. Both activators and inhibitors of STING, described below or analogs thereof, could have potential utility in the treatment of periodontitis.”

Authors’ response: Text is re-written as suggested (page 7, line 250-255).

  1. Figure 4 image quality/distortion. Recommend replacing with a better image.

Authors’ response: Figure 4 replaced in the review (page 9).

  1. Awkward phrasing and unclear description. Although molecules may act as an agonist, using the term in the verb form “agonized” is odd because a molecule does not have feelings. Further, agonists act as inhibitors/repressors, but cyclic dinucleotides act as inducers of IFNs via the STING pathway. The paper you reference report it does this by degradation of STING proteins. The way you wrote it is very confusing. You should expand these points to make it clear what you mean by agonist activity.
  • “While STING is readily agonized through the binding of cyclic dinucleotides such as c-di-GMP, c-di-AMP, 2’-3’cGAMP and 3’-3’cGAMP [60], key challenges that limit their direct utility as drug agonists include high negative charge, hydrophilicity and propensity towards degradation of cyclic dinucleotides by phosphodiesterases such as ENPP1 [61,62].”

Authors’ response: The sentence in question has been edited to better explain the role of Cyclic dinucleotides as "agonists" - which by definition refers to a chemical that binds to a receptor and activates the receptor to produce a biological response. There might be a confusion between the terms "agonist" with "antagonist" - which refers to a chemical that binds to a receptor which results in the inhibition or repression of a biological response. Quoting verbatim from the original cited article [60]: "Thus, CDNs can act as pathogen-associated molecular patterns that function as immune enhancers in the background of infections (11–16)." "CDNs were also found to be valuable tools in cancer immunotherapy:they were shown to promote cancer-specific cytotoxic T-cell activity by a mechanism that involves induction oftype I IFN in a STING-dependent manner (22, 23)". The cited article does lend credence to the point that cyclic dinucleotides result in upregulation of immune responses via STING binding and activation. However, it is also acknowledged that the title and perspective of the cited article may be inappropriate for the theme of the review and the point to be backed up. Hence, [60] has been changed to reference a citation from (Burdette, D.L.; Vence, R.E. STING and the innate immune response to nucleic acids in the cytosol. Nat Immunol, 2013, 14, 19–26) instead, which clearly explains the role of cyclic dinucleotides acting as activators or inducers for the STING pathway via STING binding and dimerization.

  • Information is included into section 6 (page 9 and 10, line 305-351).